# Fire Resistance of Phosphogypsum- and Hemp-Based Bio-Aggregate Composite with Variable Amount of Binder

**Girts Bumanis** [1] **, Martins Andzs** [2] **, Maris Sinka** [3] **and Diana Bajare** [1,*]

1. Department of Building Materials and Products, Riga Technical University, LV-1658 Riga, Latvia
2. Laboratory of Biomass Eco-Efficient Conversion, Latvian State Institute of Wood Chemistry, Dzerbenes St. 27, LV-1006 Riga, Latvia
3. 3D Concrete Printing Laboratory, Faculty of Civil Engineering, Riga Technical University, LV-1048 Riga, Latvia
* Correspondence: diana.bajare@rtu.lv

**Abstract:** Bio-aggregate composites (BACs) are typically formed by binding plant origin aggregates using organic or inorganic binders. Composite boards are being manufactured from hemp shives and Portland cement or lime and such material is associated with the so-called "hempcrete". To reach a low greenhouse gas emission rate, alternative binders must be considered. Gypsum binder releases a seven times lower amount of $CO_2$ during production compared with Portland cement, while waste gypsum can be even more efficient. In this research, gypsum-based BACs were elaborated and tested. Phosphogypsum was evaluated as an alternative binder. The objective of the research was to evaluate the fire resistance of gypsum- and phosphogypsum-binder-based BAC. In this study, the amount of binder was varied and BACs with a density from 200 to 400 kg/m$^3$ were tested. For the first time, commercial gypsum- and phosphogypsum-based hemp shive BAC fire performance was evaluated using a cone calorimeter. Results indicate that the role of gypsum content has a significant effect on the fire resistance. Time on ignition increased from 14 to 19 s and the heat release rate peak was reduced by 57%. Phosphogypsum binder, compared with commercial gypsum, showed a slight improvement of fire resistance as impurities with high water attraction are in the structure of PG.

**Keywords:** hempcrete; gypsum; phosphogypsum; fire performance; cone calorimeter

## 1. Introduction

Generally, fibers derived from plant by-products are abundant and inexpensive, mostly biodegradable, and have a low density [1]. Insulation materials made of bio-aggregate represent a significant potential to eliminate the environmental burden in relation to the use of building materials [2]. The most common bio-aggregates derived from plants used to produce insulation materials are straws, hemp shives, wood chips and sawdust. Other potential bio-materials are extensively researched. The latest reports have been published about seagrass fiber use in production of gypsum composites [3]. One of the primary problems related to the use of bio-composites is the flammability. Supplementary chemical admixtures such as boric acid must be incorporated in the mixture composition to improve materials fire performance [4]. Borate group of chemicals are banned from selling in EU and further restriction of its use is considered. The combustion performance of thermal insulation materials is affected by their own properties, thickness, surface area, fire size, fire location, and more [5]. Flammable and combustible building insulation materials have become a new class of fire hazard. This is why, historically, the application of combustible materials has been limited in use in high-story buildings [6]. Content of cellulose, hemicellulose, and lignin in bio-composites effects combustion, and correlation to lignin content has been reported for combustion and pyrolysis results [7].

In the Euroclass system, building products are divided into seven classes on the basis of their reaction-to-fire properties. The performance description and the fire scenario for each class are presented, and the highest possible European class for fire-retardant wood

products is class A2 while low-density fiberboards are assigned class E [8]. Meanwhile, most natural wood products obtain the European class D-s2,d0 with known and stable fire performance [9]. The challenge is to develop a composite that will not burn and will maintain its level of mechanical performance [10].

The incorporation of flame retardant in the system such as polylactic acid, phosphorous agents, and diammonium phosphate is an option to improve fire resistance. Flame retardant in the system decreases the peak of heat release rate (pHRR), slowing down damage caused by fire [11]. The use of fire retardant can bring drawbacks such as the adhesion of the fiber and the retardant can be unevenly distributed in the material matrix [10]. The vulnerability to fire can be partially solved with the use of an inorganic matrix which can resist temperatures up to 1000 °C but provides protection to biomaterial for a short duration [10].

In regard to the thermal properties of materials, the size of the fixed position of the ignition source has great influence on the combustion characteristics of materials. Among the fire response tests by the existing materials, cone calorimeter testing has become a more internationally recognized standard that can simulate the heat combustion behavior of materials under real fire conditions to a certain extent [5]. Cone calorimetry is one of the most effective medium-sized polymer fire behavior tests and data obtained from the test is widely described in the literature [12]. The cone calorimeter uses the "Oxygen Consumption Principle" to measure the heat release rate (HRR). It is based on the fact that an X amount of heat is released per kilogram of oxygen consumed. The instrument measures the changes in oxygen and gas concentration in order to calculate the heat release rate of a material [13]. With a cone calorimeter, it is not possible to determine the reaction to fire classification as a proper evaluation method is described in EN 13501-1.

Improvement of the thermal stability and the fire behavior of bio-based materials are eagerly perused. The low-lignin-content fibers (flax and hemp) in BACs are those that have the best fire behavior [7]. The main gaseous products of the torrefied biomass combustion process are $CO_2$ and $H_2O$, which confirms that carbon and hydrogen are significant compounds in torrefied biomass [14]. Previous reports about BAC fire performance indicate that peak HRR (pHRR) was 74 $kW/m^2$ and total heat release (THR) was from 45 to 60 $MJ/m^2$ (subjected to heat flow of 25 $kW/m^2$) for hemp shives and corn-starch-based BAC boards with a density of approximately 400 $kg/m^3$ [15]. BAC with mineral binder can increase the fire performance. The pHRR, HRR, total heat release, and average mass loss rate of the cement-bonded particleboards are noticeably decreased with increasing cement–wood ratios, and the residual mass of the cement-bonded particleboards is increased by 51% [16].

Another well-known mineral binder used for fire safety solutions is gypsum. Gypsum serves as a noncombustible substrate with beneficial fire resistance properties derived from the bound water of hydration in the gypsum. The gypsum will dehydrate under fire exposure conditions. The extent of dehydration depends on the intensity and duration of the exposure. The dehydration process absorbs considerable heat while maintaining relatively low temperatures within and behind the gypsum wallboards until the dehydration is complete. Gypsum wallboards exposed to constant incident heat fluxes of 50 $kW/m^2$ showed a pHRR 111 $kW/m^2$, THR 1.561 $MJ/m^2$, and burning duration of 11 s [17]. Such a high pHRR was reached as gypsum wallboard has a paper facer that contributes significantly to the heat release. Alternative synthetic gypsum sources can be used to prepare the binder. Deeper interest has resulted in phosphogypsum (PG), as its application up to now has been limited due to different hazardous constituents in the composition. Previous researches have proven that PG has high gypsum content, and high strength gypsum binder can be obtained from such material. The most interesting aspect is that PG contains compounds such as $P_2O_5$ with powerful water attraction properties, which could be an advantage of using PG as a material for fire protection [18–20]. In combination with natural aggregates, the potential health hazard of PG can be significantly reduced as it is used in small quantities and are diluted by the bio-aggregate mixture. Even more, the natural appearance and properties of PG can be used in favor to improve fire performance of BAC.

The aim of this work was to study the fire performance and the fire behavior of experimentally prepared gypsum-based BACs. The gypsum type and its content in mixture composition was evaluated. For the first time, phosphogypsum as a binder was evaluated in preparation of BAC.

## 2. Materials and Methods

### 2.1. Gypsum-Based Binder and Hemp Shives as Bio-Aggregate

Commercial gypsum (BG) and phosphogypsum (PG) were used to produce BAC. Commercial $\beta$-CaSO$_4$·0.5H$_2$O gypsum (Knauf Ltd., Sauriesi, Latvia) was used. Dihydrate phosphogypsum (CaSO$_4$·2H$_2$O) analyzed in this work is a waste generated by the fertilizer production plant AB Lifosa (Lithuania) in wet-process phosphoric acid production. The chemical composition of BG and PG is provided in Table 1. The difference between the two gypsum types is the amount of gypsum in the composition which for BG is 89% while for PG it is 93.8%. The BG has minor additives of MgO, Al$_2$O$_3$, and SiO$_2$, while PG has some impurities such as P$_2$O$_5$ and SrO.

**Table 1.** Chemical composition of commercial gypsum BG and phosphogypsum PG.

| Component | BG | PG |
|---|---|---|
| LOI 950 °C | 22.43 | 19.24 |
| Na$_2$O | 0.31 | 0.48 |
| MgO | 3.92 | 0.21 |
| Al$_2$O$_3$ | 1.68 | 0.71 |
| SiO$_2$ | 3.73 | 1.07 |
| P$_2$O$_5$ | 0 | 0.57 |
| SO$_3$ | 30.9 | 37.38 |
| CaO | 35.64 | 37.16 |
| TiO$_2$ | 0.05 | 0.11 |
| Fe$_2$O$_3$ | 0.46 | 0.22 |
| As$_2$O$_3$ | 0.07 | 0.09 |
| SrO | 0.23 | 2.25 |
| CeO$_2$ | 0.01 | 0.24 |
| TOTAL | 99.42 | 99.74 |

The most popular hemp species grown in the Baltic sea region are Futura 75, Finola, Uso 31, and Bialobrzeskie. They are associated with cellulose content from 43.5 to 43.7%, hemicellulose content from 23.2 to 31.8%, and lignin content from 22.0 to 26.6% [21,22]. Hemp shives (HS) with a size from 2 to 40 mm were used as a natural fiber to produce gypsum-based BAC. The raw materials were previously described in details in previously published manuscripts by the authors [23,24].

### 2.2. Mixture Composition

BAC mixture compositions with different gypsum types (BG and PG) and content in mixture composition (200, 300, or 400 kg/m$^3$) were prepared. Table 2 represents the mixture composition, and the mixing procedure was described in a previously published manuscript about the biodeterioration of the same BAC samples [24]. Reduced gypsum content provides lower environmental impact, while increasing the gypsum content provides higher strength and fire protection of BAC. A self-bearing material with stable structural integrity was obtained for all mixtures.

**Table 2.** Mixture compositions of gypsum-based BAC.

| Composition | Component | | | | | |
|---|---|---|---|---|---|---|
| | HS | BG | PG | Set Retarder | W | W/B |
| B1 | | 100 | - | - | 180 | 1.90 |
| B2 | | 200 | - | - | 260 | 1.30 |
| B3 | 120 | 300 | - | - | 290 | 1.0 |
| P1 | | - | 100 | 0.3 | 180 | 1.90 |
| P2 | | - | 200 | 0.6 | 260 | 1.30 |
| P3 | | - | 300 | 0.9 | 290 | 1.0 |

*2.3. Testing Methods*

The fire behavior of BAC was studied with a cone calorimeter test according to ISO 5660-1 (Figure 1). Samples were exposed to a radiant heat flux of 50 kW·m$^2$. Test duration for all samples was 300 s. Samples were conditioned in a controlled environment (23 °C, 50% RH, and 350 h) before the test, as the moisture content of the specimen plays a significant role in the measured results. Samples with dimensions of 10 × 10 × 5 cm were cut from larger specimens and their mass was determined. The cutting exposed the bio-filler (HS), and this could reduce fire resistance of BAC as an open structure of HS could initiate self-ignition and burn with flame. The heat release rate (HRR) was measured as a function of time, and time to ignition (TTI), total heat release (THR), peak of heat release rate (pHRR), and effective heat of combustion (EHC) were determined. All experiments were repeated three times. It is commonly agreed that the accuracy of HRR values in the cone calorimeter test is approximately 15%. The dispersion of the obtained results for the same sample was within this range [7]. The testing procedure is provided in Figure 1.

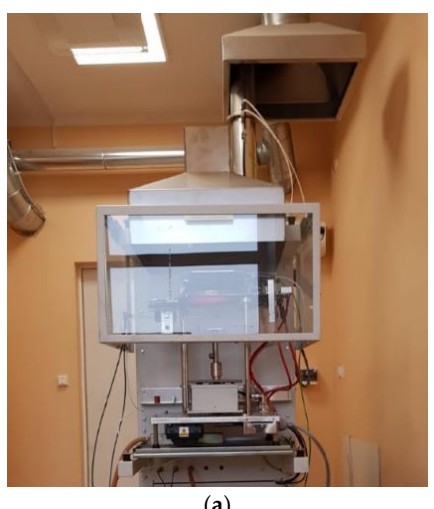
(**a**)

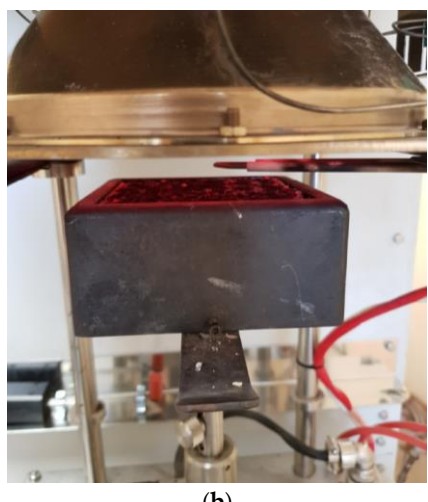
(**b**)

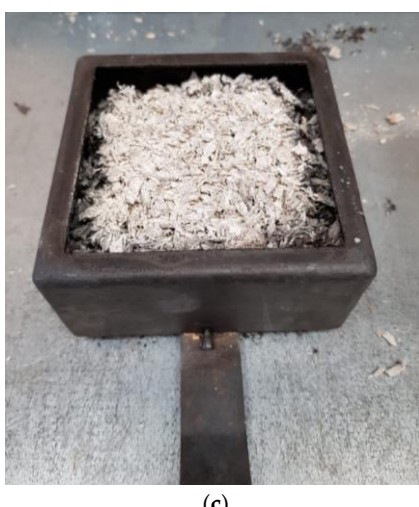
(**c**)

**Figure 1.** Experimental setup of cone calorimeter test method according to ISO 5660-1: (**a**) cone calorimeter; (**b**) test procedure; (**c**) tested sample.

## 3. Results

*3.1. Physical and Mechanical Properties*

The general properties (macrostructure, physical, and mechanical properties) of prepared BAC were described in a previous paper [24]. There are distinct differences in the macrostructure between samples with varying gypsum contents (Figure 2). In Figure 2a,b, BAC with low gypsum content shows an open structure with a small quantity of gypsum particles around hemp aggregates. Figure 2c,d show a gypsum-rich structure with embedded HS particles (B3). This led to the improved bonding of BAC, increased density (from 200 to 400 kg/m$^3$) and improved mechanical properties (0.28 to 0.57 MPa) (Table 3). The

thermal conductivity increased from 0.058 to 0.070 and 0.086 W/(m·K), respectively. The type of gypsum did not affect the mechanical and physical properties of the BAC, while GB had higher compressive strength for a binder (16 MPa) compared with PG (8 MPa). Gypsum content will show a significant effect on the reaction to fire.

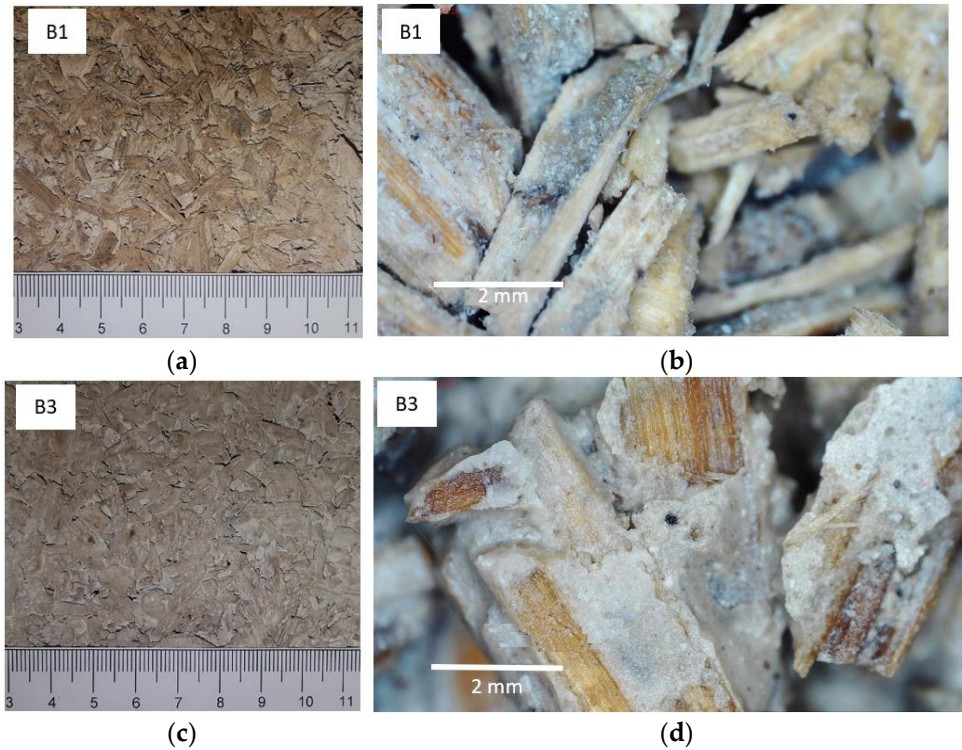

**Figure 2.** Macro and micro structure of BAC: (**a**) macro and (**b**) micro structure of BAC with low gypsum content (B1); (**c**) macro and (**d**) micro structure of BAC with high gypsum content (B3).

**Table 3.** Physical and mechanical properties of gypsum BAC.

| Mixture Composition | Density, kg/m$^3$ | Compressive Strength, MPa | Thermal Conductivity, W/(m·K) | Adsorbed Moisture, % |
|---|---|---|---|---|
| B1 | 190 ± 10 | 0.11 ± 0.05 | 0.058 | 3.8 |
| B2 | 300 ± 15 | 0.36 ± 0.05 | 0.070 | 3.6 |
| B3 | 395 ± 15 | 0.57 ± 0.08 | 0.086 | 3.5 |
| P1 | 210 ± 10 | 0.10 ± 0.03 | 0.070 | 9.3 |
| P2 | 320 ± 15 | 0.28 ± 0.08 | 0.072 | 9.2 |
| P3 | 400 ± 15 | 0.35 ± 0.08 | 0.101 | 9.2 |

Adsorbed water content for conditioned samples before the fire performance test indicates that water content was in the range from 3.5 to 3.8% for BAC with BG and from 9.2 to 9.3% for PG. An increased amount of adsorbed moisture of BAC with PG was associated with impurities present in PG. The PG could be contaminated with technological impurities such as orthophosphoric acid ($H_3PO_4$), sulfuric acid ($H_2SO_4$), calcium orthophosphate, calcium fluoride ($CaF_2$), hexafluorosilicic acid ($H_2SiF_6$), phosphates, and other rare-earth elements [25,26]. The total $P_2O_5$ content in PG traditionally is in the range from 0.28 to 1.12%, while soluble $P_2O_5$ content is in the range from 0.03–0.37% [27,28]. Additional water attraction of such substances is highly likely.

### 3.2. Reaction to Fire

The appearance of BAC with different gypsum contents after the fire test is provided in Figure 3. It can be observed that a char-forming fire-retardant veil was formed on BAC. BAC composition B1 and P1 were almost completely burned while P2 and P3 showed partial degradation of cross section under reaction to fire. For composition P1 and B1, it was almost impossible to measure the char layer as almost all material was burned due to the open structure of the material. P3 and B3 char layers reached 2 to 3 cm from the total thickness of 5 cm samples.

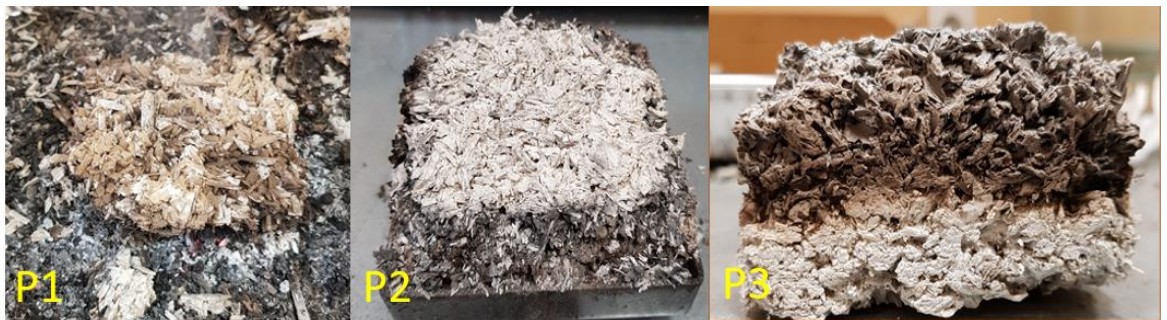

**Figure 3.** Samples after exposure to heat in the cone calorimeter test.

Time to sustained flaming is provided in Figure 4. Time to ignition increases from 14 to 19 s for BAC with BG, while with PG this time is from 15 to 19 s. Higher initial water content after conditioning of the samples did not affect the time to ignition significantly.

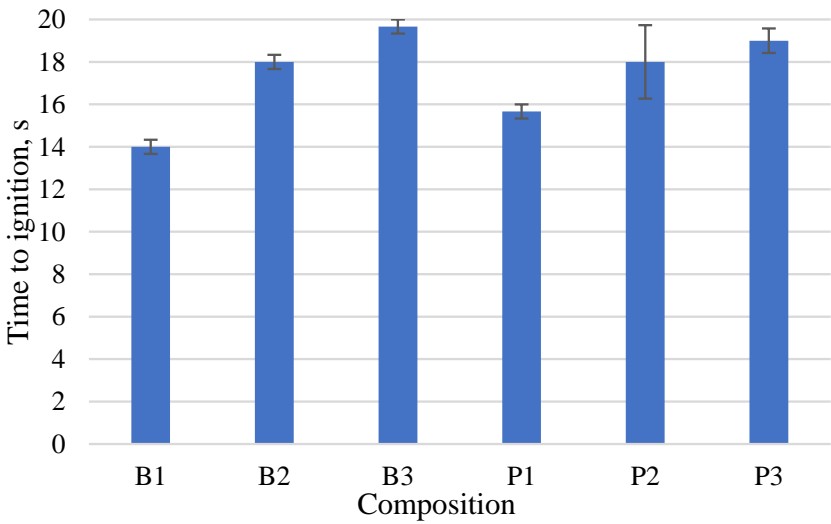

**Figure 4.** Time to ignition for BAC. B1 to B3—samples with different amounts of commercial gypsum, P1 to P3—samples with different amounts of phosphogypsum.

Loss of weight during the test is provided in Figure 5. Weight loss is associated with the evaporation of the adsorbed water (i), dehydration of gypsum (ii), and burning of hemp shives (iii). The lowest weight loss was for B3 as the highest gypsum content was for this composition and the burned cross section of the specimens was the shallowest. Furthermore, there were differences between weight loss during the fire test; significant correlation regarding the gypsum type and role of its composition was not detected. Higher impact on weight loss is associated with the open structure of BAC when individual hemp shives are subjected to fire and can burn out completely.

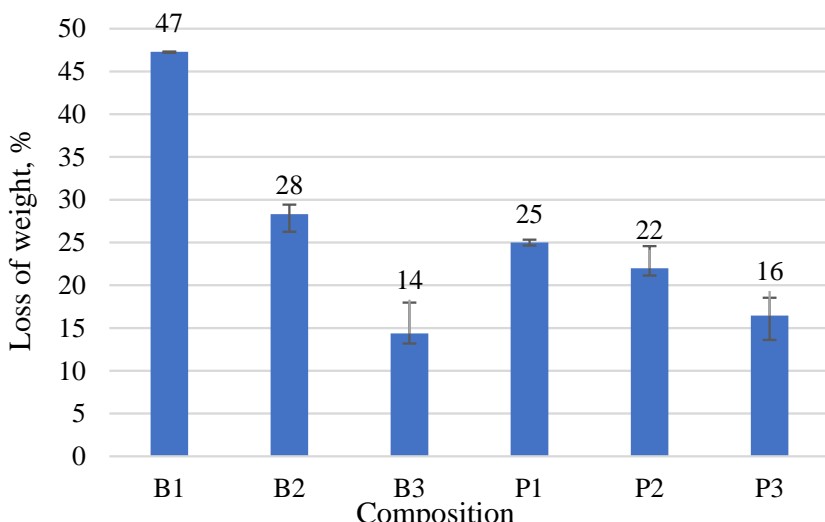

**Figure 5.** Loss of weight to initial weight. B1 to B3—samples with different amounts of commercial gypsum, P1 to P3—samples with different amounts of phosphogypsum.

Impact of external heat flux and BAC composition on heat release rate (HRR) phases of thermal degradation is different between the mixtures (Figure 6). Low gypsum content (B1 and P1) shows a sharp HRR peak at an early stage (137 kW/m$^2$ and 140 kW/m$^2$) and heat release continues for a longer time as the structure of BAC is open and individual hemp shives are poorly protected by a gypsum cover. For compositions B2 and P2, a significantly lower HRR peak was observed (80 kW/m$^2$ and 95 kW/m$^2$) and less energy was released during the test. The lowest HRR peak was for B3 and P3 (65 kW/m$^2$ and 60 kW/m$^2$). Further heat release during the test was strongly limited compared with the mixtures with lower gypsum content.

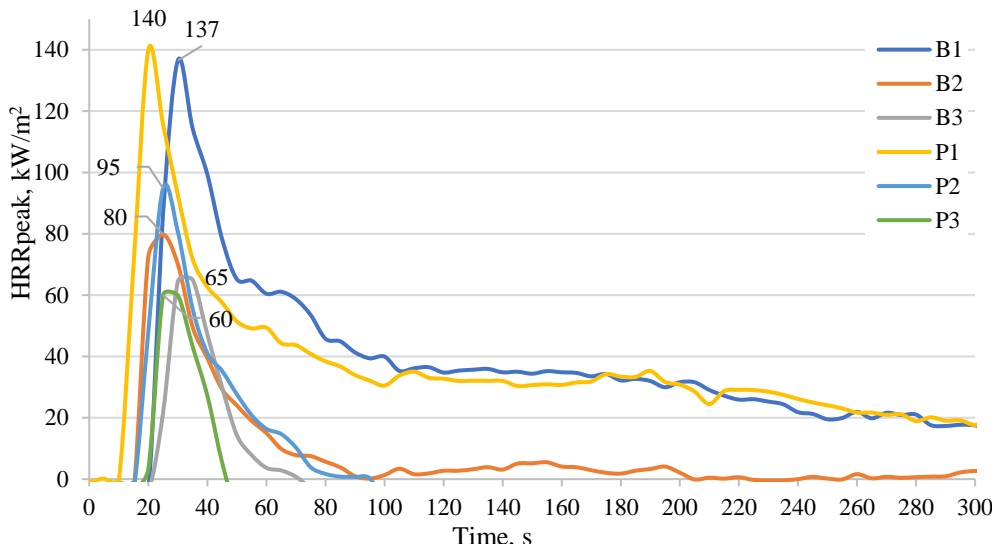

**Figure 6.** Heat release rate of BAC during the cone calorimeter test.

The maximum average heat rate of emission (MAHRE) is best thought of as an ignition-modified rate of heat emission (Figure 7). This parameter can be used in ranking materials according to their ability to support flame spread to nearby objects [29]. The reduction in MAHRE suggests significant alteration in the fire reaction behavior of the BACs with increasing gypsum content. For BACs with BG as a binder, MAHRE reduced from 57 to 24 kW/m$^2$ or by 58%, while for BAC with PG as a binder, MAHRE reduced from 64 to 23 kW/m$^2$ or by 64%.

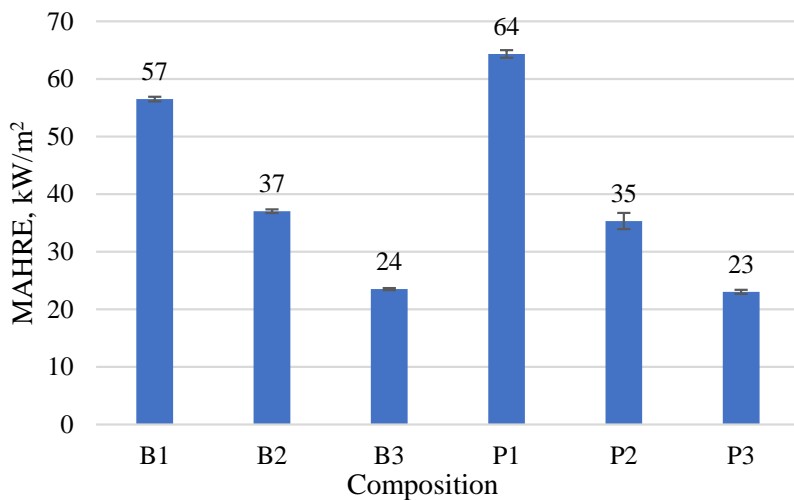

**Figure 7.** The maximum average heat-rate emission (MAHRE) of BACs.

Total heat release (THR) rate results between test time 0 and 300 s indicate the role of material density on heat release rate for 1g of material (Figure 8). Rapid burning of B1 and P1 BAC and the low density of material provides a high THR rate—1.18 kJ/g for B1 and 0.95 kJ/g for P1, respectively. This corresponds to the appearance of BAC after the test when B1 and P1 samples were completely disintegrated. A significant THR decrease was achieved by B2, B3, P2, and P3. Limited flaming and burning of BACs show a THR decrease to 0.19 kJ/g for B2 and 0.14 kJ/g for P2. B3 and P3 compositions reduced THR to 0.05 kJ/g and 0.06 kJ/g.

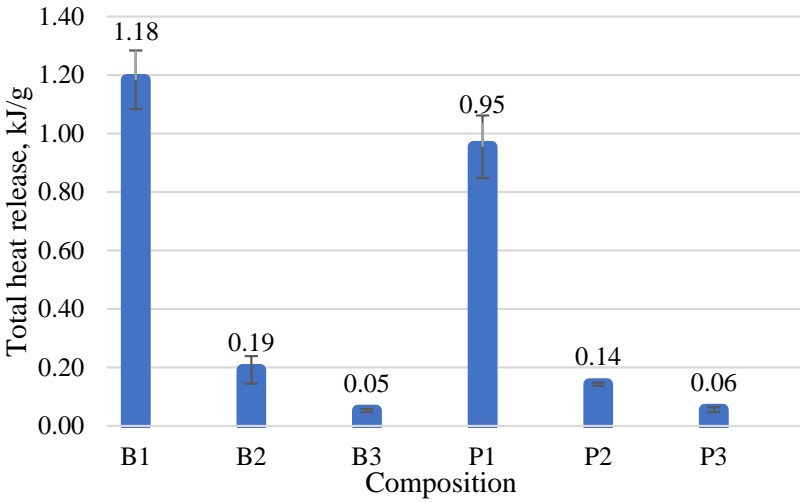

**Figure 8.** Total heat release of BAC.

## 4. Discussion

The open structure of BAC composition P1 and B1 allows heat and flame to reach the deeper layers of the material and both the surface and inner structure of the material is exposed. Low gypsum content in BAC (P1, P2 and B1, B2) ensured low protection of BAC and, in the case of P1 and B1, the material completely disintegrated during the test and it was not possible to remove it in one piece from the test frame (Figure 3).The material was not fully burned for BAC compositions B3 and P3, meaning that the limited flame spread occurs in the structure of the BAC.

It is visible that increased gypsum content increases the time needed for the start of the flaming. The results are similar for both gypsum types and higher initial adsorbed water content for PG did not play a significant role on this parameter. According to the literature,

the obtained results are similar to the painted softboard flaming time (16 s with 50 kW/m$^2$ exposure) and close to the plywood results (24 s) [30]. The open structure of BAC could support the evaporation of free water from the structure, and no fire retardation properties were observed. Results with BAC correlate with material density. Painted softboard with a density of 304 kg/m$^3$ showed a similar result with B2 and P3, while plywood with a density of 513 kg/m$^3$ showed a better performance with 24 s [30]. Time to ignition above 10s is associated with the heating up of bio-aggregates and gypsum protection as pure hemp fibers have an ignition time from 1.3–7.3 s [31].

The low-density BAC sample observation after the cone calorimeter test indicates that the temperature in the samples can reach above 400 °C when the decomposition of cellulose and degradation of lignin occur [32]. $P_2O_5$ compounds in the structure of PG, together with $H_2O$, can form $H_3PO_4$, which melts at 42.35 °C, and its solubility in water is 548 g per 100 g of water at 20 °C. When subjected to fire, decomposition products are converted to pyrophosphoric acid ($H_4P_2O_7$), when heated to 213 °C, and to metaphosphoric acid above 300 °C; final decomposition occurs at 600 °C when $P_2O_5$ is formed again [33,34].

pHRR of BAC P1 and B1 is higher compared with the data from the literature, where hemp pHRR was 108 W/g, while current results with the lowest gypsum content are close to cellulose pHRR with 141 W/g [7]. The combustion rate remained high for P1 and B1. Increased gypsum content reduced HRR significantly. With the highest gypsum content (B3 and P3), pHRR was reduced by 54–56%. Heat release for BAC with a low gypsum content continued for the whole duration of the test, while for the higher gypsum content, heat release decreased and stopped after 50 to 75 s. It is associated with the burning, with an open flame, of hemp shives and this explains the disintegration of samples P1 and B1 after the test. A more pronounced pre-initiation phase was observed for BAC with higher gypsum content. The obtained results demonstrate that BAC with high gypsum content has endothermic gypsum dehydration phenomena, which results in lower overall HRR and gas and surface temperatures. Calcium sulphate di-hydrate ($CaSO_4 \cdot 2H_2O$) loses 75% of its water, forming calcium sulphate hemi-hydrate ($CaSO_4 \cdot \frac{1}{2}H_2O$). This reaction is highly endothermic and, as a result, heat transfer through BAC is practically impeded until the dehydration process is complete. The mass loss between 50 and 150 °C is also attributed to the release of water absorbed by the fibers. Early in the fire growth stage, there is an adequate oxygen amount to mix with the heated gases, which results in flaming combustion. As the oxygen level within the structure is depleted, the fire decays, the heat release from the fire decreases, and, as a result, the temperature decreases [35]. Significant impact of heat flux on average HRR for all investigated time intervals was between 60 and 300 s. A higher adsorbed-water-content role was noticed at intervals after first ignition. It verifies that the PG binder increases the fire resistance of BAC as PG contains impurities such as phosphoric acid and its salts which produce an improved fire resistance.

MAHRE was reduced up to 58% for BAC with BG and 64% with PG. Gypsum, together with the attachment of a char-forming fire-retardant veil onto the heat-exposed surface, brought limited flame spread in the structure of the BAC. The presence of a highly consolidated surface char and gypsum inhibits the conduction of heat and/or transportation of oxygen into the pyrolysis zone, thereby reducing the net heat flux which would have otherwise increased heat emissions [29].

Results show significant THR release reduction per 1 g of material while BAC density increase was 200, 300, and 400 kg/m$^3$. Furthermore, BACs with the lowest gypsum content show poor fire performance compared with its denser counterparts; the results show heat release improvement compared with pure hemp THR rate, which is 6.2 kJ/g, as reported before [7]. This shows the important role of a gypsum binder in combination with bio-composites to improve its fire performance.

## 5. Conclusions

The obtained results show the significant role of gypsum content in the fire performance of BAC while the binder type had a minor effect. The following conclusions can be outlined:

1. Increased amount of gypsum creates a dense BAC structure, which ensures the protection of separate hemp shives, leading to the general fire performance improvement. A char layer on high-gypsum-content BAC was formed which reduced the pHRR because of the limitation of mass and thermal transfer.

2. The constituents and impurities present in PG attracted adsorbed moisture during conditioning before the test. Adsorbed water and gypsum dehydration during the test reduced heat release from the material structure. However, no significant improvement for samples with PG was detected.

3. The total HRR was significantly affected by gypsum content in the BAC. THR reduction from 1.18 kJ/g to 0.05 kJ/g was achieved. BACs with a density of 400 kg/m$^3$ and high gypsum content (300 kg/m$^3$) reduced MAHRE by 58–64% compared with low-density BACs (200 kg/m$^3$) with gypsum content of 100 kg/m$^3$.

**Author Contributions:** Conceptualization, G.B., M.S. and D.B.; methodology, G.B. and M.A.; software, M.A.; validation, G.B., M.A. and M.A.; formal analysis, G.B. and M.A.; investigation, G.B. and M.A.; resources, M.A. and D.B.; data curation, G.B. and M.A.; writing—original draft preparation, G.B. and M.A.; writing—review and editing, M.S. and D.B.; visualization, G.B. and M.A.; supervision, D.B.; project administration, M.S. and D.B.; funding acquisition, D.B. All authors have read and agreed to the published version of the manuscript.

**Funding:** This research was funded by European Regional Development Fund project "A new concept for low-energy eco-friendly house", Grant Agreement No 1.1.1.1/19/A/017.

**Data Availability Statement:** Data available on request due to restrictions of the project privacy.

**Conflicts of Interest:** The authors declare no conflict of interest.

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
