# Peer review of "Fire Resistance of Phosphogypsum- and Hemp-Based Bio-Aggregate Composite with Variable Amount of Binder"

_jcs, doi:10.3390/jcs7030118_

Round 1

Reviewer 1 Report

The paper proposed the use of gypsum binder instead of silicate cement to prepare bio-aggregate composites to reduce the CO2 emission during the production of silicate cement, and investigated the effect of different contents of commercial gypsum(BG) and phosphogypsum(PG) on the fire resistance of bio-aggregate composite. The study is innovative, but there are still some problems.

1. The composition of plaster binder and hemp stick should be listed in Section 2.

2. Section 2 only introduces hybrid composites of bio-based composites, and the preparation process needs to be clarified.

3. The discussion of the results in 3 is too simplistic and the structural framework is confusing, so it should be discussed separately.

4. The discussion in Line 187-190 about "No significant correlation was detected between the type of gypsum and the effect of its components" is incorrect, there is a correlation according to the results.

5. "P3 is 0.14kJ/g" in lines 216-217 is wrong.

6. The drawings in Figures 4-8 need to be improved.

7. The conclusion is more like a discussion of the results, please refine it.

Author Response

Dear reviewer,

thank you for your feedback and suggestions regarding to the improvements of this manuscript. In this answer I provide you with our latest improvements of the manuscript as in accordance with the reviews. Thank you for your time and effort. Please see the attachment.

Reviewer 2 Report

Dear authors and editors, I believe the article can be accepted for publication. The abstract needs to be tweaked a bit.

A good analytical overview of the topic in the Introduction. Goals and objectives are set. The experiment is clearly described. All data are scientifically explained and justified. The literature review is sufficient, the conclusions are consistent with previously published relevant sources. The article is well designed, logically structured, and generally relevant.

According to the abstract, a more detailed description of the results obtained (article summary) is desirable. Maybe a few references to MDPI publisher's articles would also improve the article.

Author Response

Dear reviewer,

thank you for your feedback and suggestions regarding to the improvements of this manuscript. In this answer I provide you with our latest improvements of the manuscript as in accordance with the reviews. Thank you for your time and effort.

Round 2

Reviewer 1 Report

The paper has been improved accordingly according to the revision, but there are still some problems.

1. The conclusion refinement is not concise enough, please elaborate in points.

2.The section 2.1 of the composition of the gypsum binder should be expressed in a table.

3.The plotting of Figure 4-Figure 8 still needs to be improved, such as the modification of horizontal and vertical axis scales and picture size.

Author Response

Thank you once again for additional comments and suggestions. The conclusions were revised and structured according to the importance of the research findings and also was elaborated in points. Second remark was completed as well, while the third issue about figure formatting was checked by the authors and only small adjustments were made with justifications in the answer below.

  1. The conclusion refinement is not concise enough, please elaborate in points.

The conclusions were revised and regrouped according to the importance of the findings.

Obtained results show significant role of gypsum content on fire performance of BAC while the binders type had minor effect. Following conclusions can be outlined:

  1. Increased amount of gypsum creates dense BAC structure, which ensures protection of separate hemp shives leading to the general fire performance improvement. A char layer on high gypsum content BAC was formed, reducing the pHRR because of the limitation of mass and thermal transfer.
  2. The constituents and impurities present in PG attracted adsorbed moisture during conditioning before the test. During the test, adsorbed water and gypsum dehydration reduced heat release from the material structure. However, no significant improvement for samples with PG was detected.
  3. The total HRR was significantly affected by gypsum content in the BAC. THR reduction from 1.18 kJ/g to 0.05 kJ/g was achieved. BACs with density of 400 kg/m3 and high gypsum content (300 kg/m3) reduced MAHRE by 58-64% comparing with low density BACs (200 kg/m3) with gypsum content of 100 kg/m3.

2.The section 2.1 of the composition of the gypsum binder should be expressed in a table.

The composition of the binders is given in table and repetitious data from the main text was removed.

3.The plotting of Figure 4-Figure 8 still needs to be improved, such as the modification of horizontal and vertical axis scales and picture size.

The figures are formatted in a similar plot. The figure size is equal for similar charts (except Figure 6). The same style and size for text is applied. Axis titles were introduced and result values is represented. Authors do believe that results are clearly visible with the current axis scaling. Comparison of updates figure plot can be performed following track changes mode.

Reviewer 2 Report

Dear authors, the article has become more structured. All comments are accurate.

Author Response

Thank you for the approval. Per the other reviewer remarks, the manuscript was further improved (especially section conclusions).

Round 3

Reviewer 1 Report

The author makes the correct changes to the given review and the quality of the article was greatly improved. However, there is still a small problem. The picture information in Figure 2 is not consistent with the description of the content in the section 3.1, there is no P3 in the picture, please add the corresponding picture to support the description in the text.

Author Response

Thank you for your comments and final remark. Reviewer is correct. In figure 2 only structure of B1 and B3 is given, which describes the appearance of BAC with the lowest and highest gypsum content. The caption of the figure and also cross-reference in the text was revised and corrected.

Figure 2. Macro and microstructure of BAC: a) macro and b) micro structure of BAC with low gypsum content (B1); c) macro and d) micro structure of BAC with high gypsum content (B3).

Round 4

Reviewer 1 Report

The authors have made the correct changes to the review comments and agree to accept it.